# A Derivative of Butyric Acid, the Fermentation Metabolite of *Staphylococcus epidermidis*, Inhibits the Growth of a *Staphylococcus aureus* Strain Isolated from Atopic Dermatitis Patients

**DOI:** 10.3390/toxins11060311

**Published:** 2019-05-31

**Authors:** Supitchaya Traisaeng, Deron Raymond Herr, Hsin-Jou Kao, Tsung-Hsien Chuang, Chun-Ming Huang

**Affiliations:** 1Department of Life Sciences, National Central University, Taoyuan 32001, Taiwan; supit_trai@hotmail.com; 2Department of Pharmacology, National University of Singapore, Singapore 117600, Singapore; phcdrh@nus.edu.sg; 3Department of Biomedical Sciences and Engineering, National Central University, Taoyuan 32001, Taiwan; lulu21522@yahoo.com.tw; 4Immunology Research Center, National Health Research Institutes, Zhunan, Miaoli 35053, Taiwan; thchuang@nhri.edu.tw; 5Department of Dermatology, University of California, San Diego 3525 John Hopkins Court, Rm276, San Diego, CA 92121, USA

**Keywords:** atopic dermatitis, butyric acid derivative, fermentation, microbiome, *S. aureus*

## Abstract

The microbiome is a rich source of metabolites for the development of novel drugs. Butyric acid, for example, is a short-chain fatty acid fermentation metabolite of the skin probiotic bacterium *Staphylococcus epidermidis* (*S. epidermidis*). Glycerol fermentation of *S. epidermidis* resulted in the production of butyric acid and effectively hindered the growth of a *Staphylococcus aureus* (*S. aureus*) strain isolated from skin lesions of patients with atopic dermatitis (AD) in vitro and in vivo. This approach, however, is unlikely to be therapeutically useful since butyric acid is malodorous and requires a high concentration in the mM range for growth suppression of AD *S. aureus*. A derivative of butyric acid, BA–NH–NH–BA, was synthesized by conjugation of two butyric acids to both ends of an –NH–O–NH– linker. BA–NH–NH–BA significantly lowered the concentration of butyric acid required to inhibit the growth of AD *S. aureus*. Like butyric acid, BA–NH–NH–BA functioned as a histone deacetylase (HDAC) inhibitor by inducing the acetylation of Histone H3 lysine 9 (AcH3K9) in human keratinocytes. Furthermore, BA–NH–NH–BA ameliorated AD *S. aureus*-induced production of pro-inflammatory interleukin (IL)-6 and remarkably reduced the colonization of AD *S. aureus* in mouse skin. These results describe a novel derivative of a skin microbiome fermentation metabolite that exhibits anti-inflammatory and *S. aureus* bactericidal activity.

## 1. Introduction

Skin dysbiosis has been defined as a state of microbial imbalance in the skin microbiome [1,2]. Increasing evidence indicates that probiotic bacteria in the skin microbiome can mediate fermentation [3] to rein in the overgrowth of opportunistic pathogens [4,5]. It has been well documented that the skin of patients with atopic dermatitis (AD) is more prone to the colonization and overgrowth of *Staphylococcus aureus* (*S. aureus*) [6]. AD is a chronic relapsing disease of pruritus and eczematous lesions that affects 15% to 20% of the childhood population [7]. It is a chronic, pruritic inflammatory skin disease of unknown origin that usually starts in early infancy but also affects a substantial number of adults [8,9].

*S. aureus* was found to co-exist with various skin commensal bacteria including *Propionibacterium acnes* (*P. acnes*) (now called *Cutibacterium acnes* (*C. acnes*) and *Staphylococcus epidermidis* (*S. epidermidis*) in an AD lesion [10]. *C. acnes* ferments carbohydrates into short-chain fatty acids (SCFAs) such as propionic acid. *S. epidermidis* ferments glycerol to butyric acid and acetic acid [11] that exert growth suppressive effects on USA300, a community-associated methicillin-resistant *S. aureus* (MRSA) [11]. In addition to the suppression of *S. aureus* growth, SCFAs, especially butyric acid, function as histone deacetylase (HDAC) inhibitors, thereby leading to increased acetylation of histones. Previous studies revealed that butyric acid can effectively inhibit HDAC in skin keratinocytes resulting in anti-inflammatory activity [12]. This is therapeutically relevant, since inflammatory processes are important drivers of AD. T helper type 2 (Th2) cytokines dominate in the early stage of AD, whereas a combination of Th1 and Th2 cytokines arises in chronic AD [13]. Many cytokines, such as interleukin (IL)-6, IL-5, IL-23, tumor necrosis factor alpha (TNFα), are detectable in AD lesions [13]. Functionally, previous studies have illustrated that alpha-toxin can provoke allergic skin diseases by activating mast cells and inducing both skin barrier disruption and AD-like skin inflammation. Furthermore, Toll-like receptor (TLR) 2 ligands provided by *S. aureus* promote AD through IL-4-mediated suppression of IL-10 [14]. SCFAs can regulate several immune cell functions including the production of cytokines (TNF-α, IL-2, IL-6, and IL-10). Butyric acid significantly attenuated lipopolysaccharides (LPS)-induced NFκB activation and nitric oxide production [15] and reduced IFNγ-induced IL-6 and TNF-α production in a macrophage cell line [16]. The ability of immune cells to migrate to the foci of infection can be regulated by SCFAs [17]. However, it is not clear yet how cytokines in a skin lesion of AD influence the growth of *S. aureus*. It has been known that IL-1β and IFN-γ, but not IL-6, induced a concentration-dependent increase of *S. aureus* growth [18]. Neutralization of IL-6 by monoclonal antibodies improved atopic dermatitis but was associated with bacterial superinfection [19].

Most SCFAs are malodorous and in general have short half-lives. Furthermore, a relatively high concentration (in the mM range) is necessary for a growth inhibitory effect of SCFAs toward pathogens [11]. Therapeutic levels of SCFAs in the mM range may be not feasible in vivo. In addition, high concentrations of SCFAs or their organic solvents may damage skin cells or the underlying tissues. These disadvantages present potentially insurmountable barriers that would prevent the use of native SCFAs as topical therapeutic agents. However, previous studies indicated that GW9508, an arylalkyl derivative of propionic acid, suppressed chemokine induction in keratinocytes and attenuated cutaneous inflammation at nanomolar to micromolar concentrations [20]. We have previously synthesized an esterified derivative of propionic acid which is not water soluble and has a minimum bactericidal concentration (MBC) value against USA300 of approximately 25 mM [21]. An analog of butyric acid, pivaloylomethyl butyrate (AN-9) [22], has been proposed as an anti-cancer prodrug that can produce effective concentrations of butyric acid. In the current study, with the aim to develop butyric acid analogs, a water-soluble derivative of butyric acid {N-[2-(2-Butyrylamino-ethoxy)-ethyl]-butyramide, BA–NH–NH–BA} was synthesized. The antimicrobial activity of BA–NH–NH–BA against an *S. aureus* strain that was isolated from the lesional skin of AD patients was examined.

## 2. Results

### 2.1. High Abundance of S. aureus in Lesional Skin of AD Patients

Tape strips were used to sample the skin microbiome from healthy skin and from non-lesional and lesional skin of AD patients. The bacteria on the tape strips were cultured on mannitol salt agar (MSA) plates for 3 d. As shown in Figure 1a, yellow and pink colonies formed in MSA plates. The yellow colonies were selected for 16S ribosomal RNA (rRNA) sequencing and identified as AD *S. aureus* (Appendix A). The pink colonies were recognized as non-*S. aureus* bacteria and were not sampled. Approximately 40% of all culturable bacteria from healthy skin and non-lesional skin of AD patients produced yellow colonies. By contrast, the percentage of yellow colonies detected from the tape strips collected from lesional skin of AD patients was markedly higher (>80%) (Figure 1b), indicating that the ratio of *S. aureus* to other bacteria on the lesional skin of AD patients was higher than that on either healthy skin or non-lesional skin of AD patients. This result is in agreement with previous findings of *S. aureus* overabundance in the dysbiotic skin microbiome in AD patients [23]. A single yellow bacterial colony isolated from the lesional skin of AD patients (“AD *S. aureus*”) was used for further experiments.

### 2.2. In Vitro Inhibition of AD S. aureus Growth by Glycerol Fermentation of S. epidermidis

Two experiments were conducted to determine whether the glycerol fermentation of *S. epidermidis*, a skin probiotic bacterium [24], hindered the growth of AD *S. aureus*. The first experiment, an overlay assay, was performed to examine the interference of *S. epidermidis* with AD *S. aureus* on agar plates with or without 2% glycerol (Figure 2a). *S. epidermidis* created a visible inhibitory zone against AD *S. aureus* in the presence of 2% glycerol. No inhibitory zone was detected when *S. epidermidis* and AD *S. aureus* were grown in the absence of glycerol. In the second experiment, *S. epidermidis* was co-cultured with AD *S. aureus* in media with or without glycerol. To establish an *S. aureus*-selective agar plate, the medium from the co-culture of *S. epidermidis* and AD *S. aureus* was spotted on tryptic soy broth (TSB) agar plates supplemented with 10 or 50 mM furazolidone. Furazolidone at 50 mM resulted in complete lethality of *S. epidermidis* (Appendix A) without affecting the growth of AD *S. aureus*. One day after the co-culture of *S. epidermidis* and AD *S. aureus* with or without glycerol, the medium was serially diluted and spotted on *S. aureus*-selective plates. As shown in Figure 2b,c, the concentration of AD *S. aureus* in the co-culture with glycerol (3.25 ± 0.48 × 10^6^ colony-forming unit (CFU)/mL) was significantly lower than in the co-culture without glycerol (5.25 ± 0.48 × 10^7^ CFU/mL). These results suggested that *S. epidermidis*-mediated glycerol fermentation interferes with the growth of AD *S. aureus* in vitro.

### 2.3. Inhibition of AD S. aureus Growth in Mouse Skin

To evaluate whether glycerol fermentation of *S. epidermidis* can hinder the growth of AD *S. aureus* in vivo, *S. epidermidis* and AD *S. aureus* with or without 2% glycerol were applied onto the wounded skin of Institute Cancer Research (ICR) mice for 3 d. The number of AD *S. aureus* in wounded skin applied with two bacteria in the presence of glycerol (5.25 ± 0.48 × 10^5^ CFU/mL) was approximately one log_10_ lower than that in wounded skin applied with two bacterial species in the absence of glycerol (2.25 ± 0.25 × 10^6^ CFU/mL) (Figure 3a,b). Furthermore, the level of IL-6 (Figure 3c) and the wound size (Appendix A) were markedly reduced when the wounded skin was treated with two bacterial species plus glycerol compared to the skin treated only with bacteria. The result indicated that *S. epidermidis* mediates glycerol fermentation to suppress skin colonization of AD *S. aureus* and the production of pro-inflammatory Il-6 cytokine.

### 2.4. Anti-AD S. aureus Activities of Butyric Acid and Its Derivative

In our previous publication, we demonstrated that propionic acid suppressed the growth of USA300 [21]. Since butyric acid is the only SCFA fermentation metabolites of *S. epidermidis* [25], we determined the killing activity of butyric acid against AD *S. aureus*. Butyric acid (0–500 mM) was added into the culture of AD *S. aureus* (10^6^ CFU/mL) overnight. The >1 log_10_ inhibition of butyric acid for AD *S. aureus* was 10 mM, and the concentration for complete inhibition was greater than 50 mM (Figure 4a,b). To circumvent the limitations of butyric acid’s malodor and short half-life, BA–NH–NH–BA, a water-soluble derivative of butyric acid was synthesized by conjugating two butyric acids with a –NH–O–NH– linker (Figure 5a). The anti-*S. aureus* activity of BA–NH–NH–BA was examined by adding BA–NH–NH–BA (0–500 mM) into the culture of AD *S. aureus*. The >1 log_10_ inhibition of BA–NH–NH–BA for AD *S. aureus* was 0.02 mM, while concentrations greater than 250 mM completely inhibited the growth (Figure 5b,c). The >1 log_10_ inhibition of BA–NH–NH–BA for AD *S. aureus* was 500 times lower than that of butyric acid, indicating the higher potency of BA–NH–NH–BA as an anti-*S. aureus* agent.

### 2.5. Inhibition of HDAC and Suppression of AD S. aureus Growth In Vivo by BA–NH–NH–BA

SCFAs, especially butyric acid, act as HDAC inhibitors, thereby leading to increased histone acetylation and regulation of gene transcription. To test if butyric acid and BA–NH–NH–BA act as HDAC inhibitors in skin cells, human skin HaCaT keratinocytes were treated with butyric acid, BA–NH–NH–BA, or phosphate-buffered saline (PBS), as a control, for 8 h. As shown in Figure 6a, treatment with 4 mM butyric acid or BA–NH–NH–BA significantly increased the level of acetylated Histone H3 lysine 9 (AcH3K9). To assess the effects of BA–NH–NH–BA on AD *S. aureus* growth in mouse skin, skin wounds were treated with AD *S. aureus* and with BA–NH–NH–BA (100 µM, 0.4 mM, and 4 mM) or PBS. Three days after the application, the number of AD *S. aureus* was significantly reduced by 0.4 mM BA–NH–NH–BA. The concentration of BA–NH–NH–BA at 4 mM led to >1 log_10_ suppression of AD *S. aureus* growth in skin wounds (Figure 6b,c). It has been reported that HDAC inhibition by butyric acid modulated several leukocyte functions including the production of cytokines [26]. In the Figure 6d, we found that the level of IL-6 in the AD. *S. aureus*-colonized skin was dose-dependently reduced by BA–NH–NH–BA.

## 3. Discussion

*S. aureus* is a pathogen commonly found in patients with AD. It has been reported that *S. epidermidis* can produce the phenol-soluble modulins (PSMs) γ and δ to hamper the growth of *S. aureus* [27]. Our data in Figure 3 demonstrate that *S. epidermidis* can mediate glycerol fermentation to reduce skin colonization by AD *S. aureus*. Although a high dose is required, butyric acid, a metabolite of glycerol fermentation of *S. epidermidis*, can kill AD *S. aureus* (Figure 4). In addition to *S. epidermidis*, *Staphylococcus hominis* has also been recognized as a beneficial bacterium in AD skin. *S. hominis* can produce lantibiotics which are strain-specific, highly potent, *S. aureus*-selective bactericidal agents that synergize with human antimicrobial peptides such as LL-37 [28]. Clinical studies suggest that the topical application of commensal *S. hominis* reduces AD severity [29]. Several mouse models have been developed to mimic AD in humans. Application of 2,4-dinitrocholrlbenzene (DNCB) on mouse skin shows symptoms similar to those of human AD, including epidermal hyperplasia, dermal mast cell infiltration, and elevated serum IgE levels [30]. In another model, repeated exposure of NC/Nga mice to *Dermatophagoides farinae* (*D. farinae*) crude extract (DfE) induced AD-like lesions [31]. In this study, we applied an *S. aureus* strain isolated from an AD patient onto a skin wound in mice, thus simulating AD patients with a wound on a lesional skin after persistent scratching.

Because of its short half-life, a prohibitively large dose of butyric acid may be necessary for in vivo efficacy. One method to augment the efficacy of butyric acid is to convert it into a prodrug that would generate higher concentrations of intracellular butyric acid. Prodrugs of butyric acid such as AN-9 [32], isobutyramide [33], and tributyrin [34] are lipophilic and have faster rates of intracellular penetration and/or slower rates of metabolic degradation. Data in our previous publication have demonstrated that propionic acid killed USA300 by reducing its intracellular pH [21]. The 2-(2-propionyloxyethoxy) ethylester, a prodrug of propionic acid, and acetic acid 2-(2-acetyloxyethoxy) ethylester, a prodrug of acetic acid, effectively suppressed the growth of USA300 [21] and *Candida parapsilosis* [35], respectively. At least 10 mM butyric acid is required to cause greater than 1 log_10_ inhibition in the growth of AD *S. aureus* in vitro (Figure 4). Although the mechanism of action of BA–NH–NH–BA against AD *S. aureus* is not clear, the >1 log_10_ inhibition by BA–NH–NH–BA of AD *S. aureus* was 20 µM (Figure 5). Thus, BA–NH–NH–BA dramatically reduced the required dose of butyric acid for killing AD *S. aureus* in vitro. BA–NH–NH–BA was synthesized by conjugating two butyric acids to a non-cleavable –NH–O–NH– linker. It is a water-soluble compound with a molecular weight of 244.33. As shown in Appendix A, BA–NH–NH–BA did not cause significant cell death when it was applied onto mouse skin. Furthermore, results from gas chromatography (GC) analysis (Appendix A) indicated that a 4 mM BA–NH–NH–BA solution can be stored at 4 °C for six months without degradation. Like the antibacterial activity of water-soluble and hydrophilic chitosan [36], the anti-*S. aureus* activity of BA–NH–NH–BA may be the result of changes in the properties of plasma membrane permeability, which provoke internal osmotic imbalance, consequently inhibiting the growth of bacteria. It has been reported that AD *S. aureus* and other *S. aureus* strains have different characteristics including distinct activities of clumping factor B [37] and T cell responses [38]. Our future works will determine if BA–NH–NH–BA can selectively suppress the growth of AD *S. aureus* without affecting other skin commensal bacteria.

When skin wounds were treated with AD *S. aureus* and BA–NH–NH–BA for three days, both the number of AD *S. aureus* and the level of IL-6 in the skin were attenuated relative to wounds that received *S. aureus* alone (Figure 6b–d). To mimic the over-growth of *S. aureus* in AD lesions, future works will include the inoculation of AD *S. aureus* onto the skin for few days before topical application of BA–NH–NH–BA. To rule out the possibility of reduction of IL-6 due to the elimination of AD *S. aureus* by BA–NH–NH–BA, we applied heat-killed *S. aureus* with BA–NH–NH–BA to KERTr cells. As shown in Appendix A, BA–NH–NH–BA attenuated the level of IL-6 induced by heat-killed *S. aureus*. These data suggest that BA–NH–NH–BA may directly down-regulate the production of IL-6 via the inhibition of HDACs (Figure 6a). Butyric acid inhibits most HDACs except class III HDACs and class II HDAC-6 and -10 [39]. The study of the interaction of an HDAC-like enzyme with trichostatin A (TSA), a broad-spectrum HDAC inhibitor, revealed that the aliphatic chain of TSA occupies a hydrophobic cleft on the surface of HDAC-like enzymes [40]. It is possible that butyric acid and BA–NH–NH–BA also occupy this same hydrophobic pocket to inhibit HDAC activity.

HDAC inhibition may affect immune responses to bacterial infection. Suppression of TLR-induced cytokine production by HDAC inhibition may influence the quality of immune responses to pathogens. Reduction of the production of pro-inflammatory cytokines retards neutrophil recruitment to bacteria-infected sites [41]. It has been documented that phagocytosis of *S. aureus* by immune cells was reduced by HDAC inhibition [42], suggesting that the therapeutic application of HDAC inhibitors might significantly compromise the immune system and render patients more susceptible to *S. aureus* infections. It has been reported that inhibition of HDAC8 and HDAC9 by microbial SCFAs disrupted the tolerance of skin cells to TLR ligands [43], indicating that inhibition of HDACs by SCFAs may alter skin responses to damage-associated molecular patterns (DAMPs) and pathogen-associated molecular patterns (PAMPs). HDACs in mice were depleted by 3,3’-diindolylmethane (DIM) to examine if HDAC inhibition influenced skin colonization by AD *S. aureus*. As shown in Appendix A, depletion of HDACs lowered the level of IL-6 in skin but had no effect on skin colonization of AD *S. aureus*. This result indicates that a decrease in IL-6 production after HDAC inhibition by BA–NH–NH–BA is not sufficient to change skin colonization by AD *S. aureus* (Figure 6c). The reduction of skin colonization of AD *S. aureus* by BA–NH–NH–BA, therefore, is likely to be mainly due to its bactericidal activity. Cumulatively, our study introduces BA–NH–NH–BA as a derivative of butyric acid that, like butyric acid, exerts the activity of HDAC inhibition but has higher potency than butyric acid in terms of suppression of AD *S. aureus* growth.

## 4. Materials and Methods

### 4.1. Ethics Statement

This research was carried out in strict accordance with an approved Institutional Animal Care and Use Committee (IACUC) protocol at National Central University (NCU), Taiwan (NCU-106-015, December 19, 2017). The Institutional Review Board (IRB) at Landseed hospital in Taiwan approved the consent and bacterial sampling procedure under an approved protocol (No. 16018C0, June 15, 2018). Written consent was obtained from all participants before conducting bacterial sampling.

### 4.2. Bacterial Culture

*S. aureus* bacteria were isolated from AD patients of Landseed Hospital, Taiwan. The isolated *S. aureus* (“AD *S. aureus*”) was validated by 16S rRNA sequencing using the 16S rRNA 27F and 534R primers [25] (Appendix A). AD *S. aureus* and *S. epidermidis* (ATCC 12228) were cultured in TSB (Sigma, St. Louis, MO, USA) overnight at 37 °C. The cultures were diluted 1:100 and cultured to an optical density 600 nm (OD_600_) = 1.0. Bacteria were harvested by centrifugation at 5000× *g* for 10 min, washed with PBS, and suspended in PBS for further experiments.

### 4.3. Bacterial Sampling with Tape Strips

Bacteria on the skin area of each subject were collected via tape strips. A medical air-permeable tape with acrylic glue (2 × 3 cm, 3M, St. Paul, MN, USA) was sterilized by ultraviolet radiation and thoroughly applied to the skin of healthy subjects (*n* = 3), and to non-lesional (*n* = 3) and lesional (*n* = 3) skin of AD patients. The sterilized tapes were applied to each skin area of each subject for 1 min. The tapes were then peeled off from the skin with sterile forceps. The bacteria of the skin surface adhering on tapes were transferred to MSA (Merck, Billerica, MA, USA) plates and incubated for 72 h. Bacterial colonies appeared yellow or pink in color on MSA plates. Pink colonies were referred to as non-*S. aureus* bacteria. AD *S. aureus* bacteria appearing yellow in color were confirmed by 16S rRNA sequencing.

### 4.4. Co-culture of AD S. aureus and S. epidermidis In Vitro

An overlay assay was used for the detection of glycerol fermentation of *S. epidermidis* against AD *S. aureus.* Plates contained 1.5% molten (*w*/*v*) agar (Oxoid. Ltd., London, UK) with or without 2% glycerol in TSB. Bacteria (500 µL PBS containing AD *S. aureus* 10^7^ CFU) were poured into the plates to produce a homogeneous lawn of AD *S. aureus*. Afterward, PBS (10 µL) containing *S. epidermidis* (10^8^ CFU) was spotted on top of the lawn of AD *S. aureus*, and the bacteria were then cultured at 37 °C for 2 d. An inhibition/clear zone appeared on the surface of the agar when *S. epidermidis* interfered with the growth of AD *S. aureus*. For co-culture experiments, *S. epidermidis* (10^8^ CFU/mL) and AD *S. aureus* (10^5^ CFU/mL) were cultured in 5 mL of TSB with or without 2% glycerol at 37 °C for 24 h. CFUs of AD *S. aureus* were enumerated by plating serial dilutions (1:10^0^–1:10^5^) of the co-culture medium on a furazolidone (50 mM)-supplemented TSB plate. Plates were incubated overnight at 37 °C to count the colonies.

### 4.5. In Vivo Effects of S. epidermidis Glycerol Fermentation on Skin Colonization of AD S. aureus

ICR mice (8–12-month-old females; National Laboratory Animal Center, Taiwan) were anesthetized by isoflurane. A 1 cm wound was made on the dorsal skin following shaving with electrical clippers. Following skin wounding, 10 µL PBS of AD *S. aureus* (10^7^ CFU) and *S. epidermidis* (10^7^ CFU) in the presence or absence of 2% glycerol was applied onto the wounds for 3 d. To measure the extent of wound closure, transparent parafilm was placed over the wounded skin, and the area was marked by outlining the area of the wound. The lesion size (cm^2^) was measured daily then calculated with ImageJ software [National Institutes of Health (NIH), Bethesda, MD, USA]. To determine bacterial counts, the skin wound was excised 3 d following bacterial application. The excised skin (20 mg) was homogenized in 200 µL of sterile PBS with a tissue grinder. Bacterial CFUs in the skin were enumerated by plating serial dilutions (1:10^0^–1:10^5^) of the homogenate on a furazolidone-supplemented TSB plate. The plates were incubated overnight at 37 °C to count the colonies. The bacterial number (CFUs/mL) in excised skin was calculated. The pro-inflammatory IL-6 cytokine was determined by sandwich enzyme-linked immunosorbent assay (ELISA) using IL-6 ELISA kits (R&D systems, Minneapolis, MN, USA).

### 4.6. Synthesis of BA–NH–NH–BA

Two butyric acids (Sigma-Aldrich, St. Louis, MO, USA) were conjugated to both ends of an –NH–O–NH– linker. Butyric acid (50 mmol) and –NH–O–NH– (20 mmol) in dichloromethane (100 mL) were added to N,N’-dicyclohexyl carbodimide (DCC) (60 mmol) portion-wise. The cloudy white suspensions were stirred at room temperature overnight, then filtered and washed with hexanes. The filtrate was concentrated under reduced pressure to yield pure (>97%) and colorless BA–NH–NH–BA which was further purified by silica gel chromatography eluting with 10% ethyl ethanoate (EtOAc)/hexanes. The conjugation of BA–NH–NH–BA was validated by ^1^H nuclear magnetic resonance (NMR) (300 MHz) analysis (Bruker DPX-300, Billerica, MA, USA) using CDCl_3_ as a solvent. To examine the anti-*S. aureus* activity, BA–NH–NH–BA was dissolved in PBS. Skin wounds of ICR mice were topically treated with 10 µL of AD *S. aureus* (10^8^ CFU) along with BA–NH–NH–BA (0.1, 0.4 or 4 mM) for 3 d. Application of 10 µL of AD *S. aureus* (10^8^ CFU) with PBS served as a control. The number of AD *S. aureus* and the level of IL-6 in skin wound were determined as described above.

### 4.7. Suppresion of Bacterial Growth

AD *S. aureus* (10^6^ CFU/mL) was incubated with butyric acid or BA–NH–NH–BA at various concentrations in PBS as indicated, in media on a 96-well microtiter plate (100 μL per well) for 24 h. The bacteria were incubated with PBS alone as a control. After incubation, the bacteria were diluted 1:10^0^–1:10^5^ with PBS. The number of bacteria was determined by spotting the dilution (5 μL) on an agar plate supplemented with medium for the counting of CFUs.

### 4.8. Cell Culture

Human HaCaT keratinocyte cells were cultured in Dulbecco’s modified essential medium (Gibco-BRL, Grand Island, NY, USA) with 10% (*v*/*v*) fetal bovine serum (Irvine Scientific, Santa Ana, CA, USA), 100 units/mL penicillin, and 100 µg/mL streptomycin. The cells (5 × 10^4^ cells/mL) were incubated for 3 d before treatment with PBS, 4 mM butyric acid, or BA–NH–NH–BA for 8 h. The cells were then harvested for detection of AcH3K9 levels by western blot.

### 4.9. Western Blotting

Lysates (30 µg) of HaCaT cells were separated by sodium dodecyl sulfate-polyacrylamide gel electrophoresis and then transferred onto a nitrocellulose membrane by use of a transfer cell (Bio-Rad, Hercules, CA, USA). Western blotting was carried out by sequential incubation in 5% non-fat milk blocking buffer at room temperature for 60 min, followed by incubation with primary antibodies against either AcH3K9 (Abcam, Cambridge, MA, USA) or glyceraldehyde 3-phosphate dehydrogenase (GAPDH) (Abcam, Cambridge, MA, USA) at 4 °C overnight, and finally horseradish peroxidase-conjugated anti-rabbit secondary antibodies (Abcam, Cambridge, MA, USA) at room temperature for 90 min. Immunoreactive bands were detected by reaction with the enhanced chemiluminescence (ECL) detection system reagent (Amersham, Arlington Heights, IL, USA).

### 4.10. Statistical Analysis

To determine significance between groups, comparisons were made using the two-tailed Student’s *t*-test. For in vivo experiments, at least three mice per group per experiment were used. Data represent the mean ± SE from three independent experiments. For all statistical tests, *p*-values of < 0.05 (*), < 0.01 (**), and < 0.001 (***) were accepted for statistical significance.

## Figures and Tables

**Figure 1 toxins-11-00311-f001:**
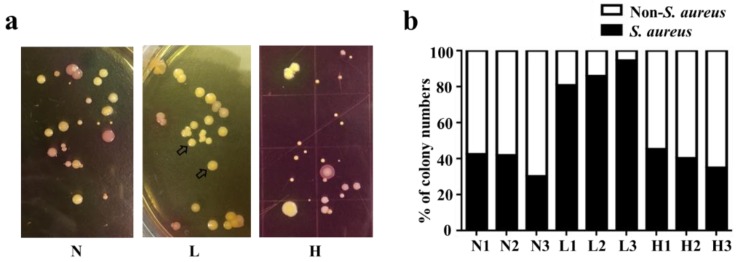
The colonization of *S. aureus* in healthy skin and non-lesional and lesional skin of atopic dermatitis (AD) patients. (**a**) Tape stripping was conducted on healthy skin (H), non-lesional AD skin (N), and lesional AD skin (L). After stripping, the pieces of tape strip were placed, adhesive side down, on mannitol salt agar (MSA) plates for bacterial growth. The colonies of *S. aureus* (arrows, yellow) and non-*S. aureus* (pink) were detected on the MSA plates. A representative picture of three independent experiments is shown. (**b**) The percentage of *S. aureus* and non-*S. aureus* colonies from three subjects with healthy skin (H1, H2, and H3) and AD patients with non-lesional (N1, N2, and N3) and lesional skin (L1, L2, and L3) are shown.

**Figure 2 toxins-11-00311-f002:**
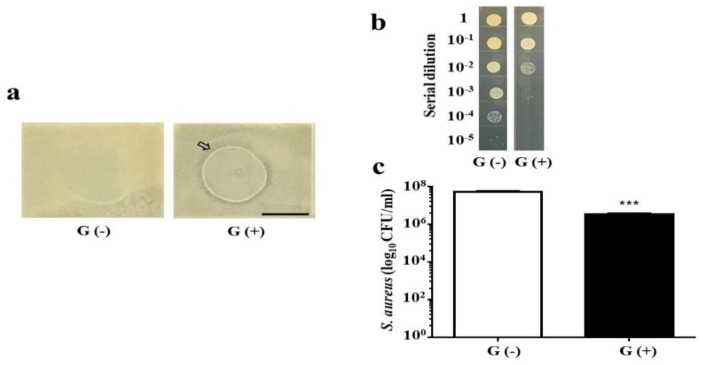
The interference of glycerol fermentation of *S. epidermidis* with the growth of AD *S. aureus* in vitro. (**a**) *S. epidermidis* (10^8^ colony-forming unit, CFU) was spotted on top of a lawn of AD *S. aureus* in the absence (G−) or presence (G+) of 2% glycerol for 2 d. An inhibition/clear zone (arrow) was observed when *S. epidermidis* was spotted on the lawn of AD *S. aureus* in the presence of glycerol. Scale bar: 1 cm. (**b**) CFUs of AD *S. aureus* were enumerated by plating serial dilutions (1:10^1^–1:10^5^) of medium from co-cultures of *S. epidermidis* (10^8^ CFU/mL) and AD *S. aureus* (10^5^ CFU/mL) with (G+) or without (G−) 2% glycerol on a furazolidone (50 mM)-supplemented tryptic soy broth (TSB) plate for 24 h. (**c**) Quantification of AD *S. aureus* from the co-cultures. *** *p* < 0.001 (two-tailed *t*-test). Data shown represent the mean ± standard error (SE) of experiments performed in triplicate.

**Figure 3 toxins-11-00311-f003:**
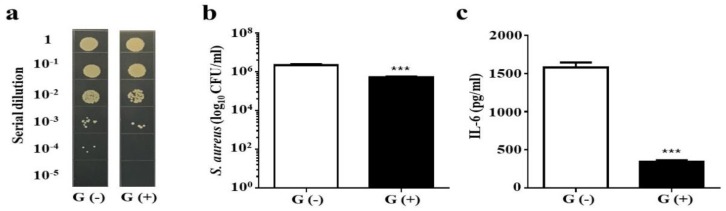
The effect of glycerol fermentation of *S. epidermidis* on the growth of AD *S. aureus* in vivo. (**a**) In the experiment, 10 µL of *S. epidermidis* (10^7^ CFU) and AD *S. aureus* (10^7^ CFU) with (G+) or without (G−) 2% glycerol was applied onto a skin wound on the dorsal skin of Institute Cancer Research (ICR) mice for 3 d. Bacterial CFUs in the skin wounds were enumerated by plating serial dilutions (1:10^1^–1:10^5^) of the skin homogenate on a furazolidone (50 mM)-supplemented TSB plate. The number (log_10_ CFU/mL) of AD *S. aureus* (**b**) and the level of pro-inflammatory IL-6 cytokine (**c**) were quantified. Data shown are the mean ± SE. *** *p* < 0.001 (two-tailed *t*-tests).

**Figure 4 toxins-11-00311-f004:**
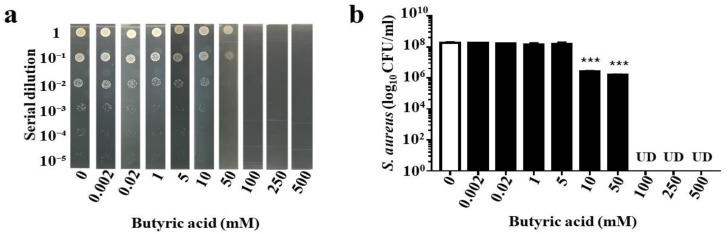
Suppression of AD *S. aureus* growth by butyric acid. AD *S. aureus* (10^6^ CFU/mL) was incubated with butyric acid (**a**,**b**) (0.002–500 mM in PBS) for 24 h. Incubation of AD *S. aureus* with PBS served as a control (0). After incubation, AD *S. aureus* was diluted 1:10^0^–1:10^5^ with PBS, and 5 μL of the dilutions was spotted on an agar plate (**a**). The CFU counts are illustrated as the mean ± SE of three independent experiments (**b**). *** *p* < 0.001 (two-tailed *t*-test). UD, undetectable.

**Figure 5 toxins-11-00311-f005:**
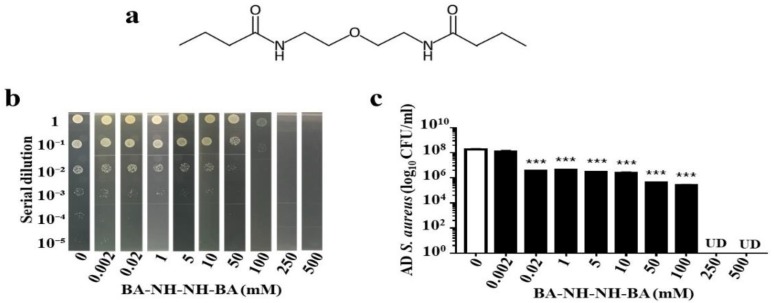
Suppression of AD *S. aureus* growth by BA–NH–NH–BA. (**a**) Chemical structure of BA–NH–NH–BA. (**b**) AD *S. aureus* (10^6^ CFU/mL) was incubated with BA–NH–NH–BA (0–500 mM in PBS) for 24 h. After serial dilutions (1:10^0^–1:10^5^), colonies of AD *S. aureus* incubated with or without BA–NH–NH–BA were grown on agar plates, and the number (log_10_ CFU/mL) of bacteria was determined (**c**). Data are the mean ± SE of three individual experiments. ** *p* < 0.01, *** *p* < 0.001 (two-tailed *t*-tests). UD, undetectable.

**Figure 6 toxins-11-00311-f006:**
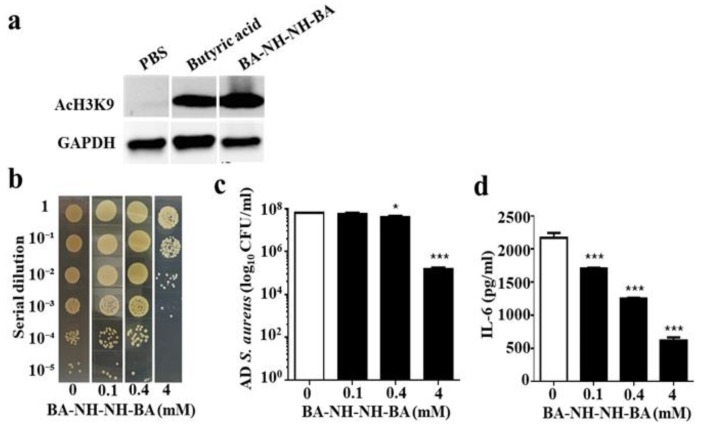
Induction of histone H3 lysine 9 acetylation (AcH3k9) and reduction of AD *S. aureus* growth and IL-6 production by BA–NH–NH–BA. (**a**) Human keratinocyte cells (HaCaT) were treated with 4 mM butyric acid or BA–NH–NH–BA in PBS for 8 h. Cells treated with PBS served as a control. The presence of AcH3K9 in cells was detected by Western blotting using an anti-AcH3K9 antibody. A representative result from three similar experiments is shown. (**b**) A 1 cm wound was made on the dorsal skin of ICR mice before applying AD *S. aureus* with PBS alone (0 mM) or BA–NH–NH–BA (0.1-4 mM in PBS) for 3 d. Bacterial CFUs in the skin wounds were enumerated by plating serial dilutions of the homogenate on a plate. The number (log_10_ CFU/mL) of AD *S. aureus* bacteria (**c**) and the level of IL-6 pro-inflammatory cytokine (**d**) were determined. Data are the mean ± SE of three separate experiments. * *p* < 0.05, *** *p* < 0.001 (two-tailed *t*-tests).

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
