# Peer review of "A Derivative of Butyric Acid, the Fermentation Metabolite of Staphylococcus epidermidis, Inhibits the Growth of a Staphylococcus aureus Strain Isolated from Atopic Dermatitis Patients"

_toxins, 2019, doi:10.3390/toxins11060311_

Round 1

Reviewer 1 Report

This manuscript is clear and well written.

I have a few questions and comments as follows:

1.      What is the main purpose of this study?

2.      Page2, line 46 and 52, some words were capital letter and please check them.

3.      Page 2, line 64 please change “topical therapeutic agents” instead of “topically applied therapeutic agents”.

4.      Page 8, line 271, please add reference of S.aureus 16S rRNA primers for confirmation of isolates..

5.      Page8, line 283, “AD S.aureus…………………….validated by 16S rRNA sequencing”. Better to use “confirmed” instead of “validated”.

6.      What do you mean ECL detection system reagents? Please briefly explain in the text.

7.      Supplementary Figure S1 is not necessary.

8.      All the abbreviations should be mentioned under material and method section.

9.      All the references should be written under reference section. Please move reference of supplementary material to reference section of main text.

10.  Page 9, line 317, what do you mean by H nuclear magnetic resonance analysis using CDCl3 solvent?

11.  What about the stability of your newly synthesized compound, BA-NH-NH-BA?

Author Response

Summary

As reviewer requested, two new experiments in Figures S6 and   S7 have been conducted to demonstrate no cytotoxicity and high stability of   BA-NH-NH-BA. Furthermore, two new references [37, 38] were added.

Reviewer 1

Comment 1:

What is the main purpose of this study?

Response 1:

The main purpose of this study   to develop butyric acid analogs has been added.

“In the current study with an aim to develop butyric acid   analogs, a water soluble derivative of butyric acid   {N-[2-(2-Butyrylamino-ethoxy)-ethyl]-butyramide; BA-NH-NH-BA} has been   synthesized.”

Comment 2:

Page2, line 46 and 52, some words were   capital letter and please check them.

Response 2:

Errors at these words have been   corrected.

Comment 3:

Page 2, line 64 please change “topical therapeutic agents” instead of “topically   applied therapeutic agents”.

Response 3:

“Topically applied therapeutic   agents” have been changed to “topical therapeutic agents”.

Comment 4:

Page 8, line 271, please add reference of S.   aureus 16S rRNA primers for confirmation of isolates

Response 4:

A reference [25] has   been added.

25. Wang, Y., et al., Staphylococcus   epidermidis in the human skin microbiome mediates fermentation to inhibit   the growth of Propionibacterium acnes: implications of probiotics in   acne vulgaris. Applied microbiology and biotechnology, 2014. 98(1): p.   411-424.

Comment 5:

Page8, line 283, “AD S. aureus…………………….validated   by 16S rRNA sequencing”. Better to use “confirmed” instead of “validated”.

Response 5:

“Validated” has been changed to   “confirmed”.

Comment 6:

What do you mean ECL detection system   reagents? Please briefly explain in the text.

Response 6:

ECL for “enhanced   chemiluminescence” has been spelled out.

Comment 7:

Supplementary Figure S1 is not necessary.

Response 7:

16S rRNA sequences have been   kept for readers to compare other S.   aureus strains they may have.

Comment 8:

 All   the abbreviations should be mentioned under material and method section.

Response 8:

All abbreviations have been   mentioned under Materials and Methods.

Comment 9:

All the references should be written under   reference section. Please move reference of supplementary material to   reference section of main text.

Response 9:

We have moved a Reference of   Supplementary Materials to the Reference section of main text.

Comment 10:

Page 9, line 317, what do you mean by H   nuclear magnetic resonance analysis using CDCl3 solvent?

Response 10:

The sentence has been   corrected.

The conjugate of BA-NH-NH-BA   was validated by 1H   nuclear   magnetic resonance (NMR) (300 MHz) analysis (Bruker DPX-300, Billerica, MA, USA) using   CDCl3 solvent”.

Comment 11:

What about the stability of your newly   synthesized compound, BA-NH-NH-BA?

Response 11:

A new experiment in Figure S7   has been conducted to demonstrate that BA-NH-NH-BA can be stable for at least   6 months.

“2.3. GC Analysis

BA-NH-NH-BA (4 mM) was dissolved in   PBS and stored at 4 for six months and detected by ethyl acetate liquid-liquid extraction   and saturation with sodium chloride followed by GC analysis using an Agilent   5890 Series II GC [11].

Figure S7. Stability of   BA-NH-NH-BA by GC analysis. BA-NH-NH-BA was detected by GC after fresh   preparation (a) six months of storage (b) at 4.  BA-NH-NH-BA with a retention time of 22.1   min (arrows) was detected.”

Reviewer 2 Report

This manuscript describes the isolation of a S. aureus strain from an atopic dermatitis lesion and the effect of BA-NH-NH-BA on S. aureus growth, IL-6 production and histone acetylation.

The effect of butyrate on S. aureus growth, IL-6 production and histone acetylation has been described before in the literature, as well as the association of S. aureus with AD. The novelty of this manuscript lies in the generation of a water-soluble derivative of butyrate, BA-NH-NH-BA. The authors suggest that this derivative is more active than butyrate and might be useful as a therapy against AD.

The first three figures show data that confirm results from previous studies (production of butyrate from glycerol by S. epidermidis, effect on IL-6 levels and growth suppression of S. aureus).

The section around figures 4 and 5 is confusing. The authors describe “complete inhibition”, but what they actually test is the minimal bactericidal concentration (MBC) which is lower than the MBC. The MBC is defined as the concentration that kills at least 99.9% of a 106/ml CFU culture. In the method section and fig 4, it says 106/ml CFU were used, fig 5 says 105/ml CFU and the Y-axes in both figures indicate 108/ml CFU as starting point. Either way, a one to two log reduction is not equivalent to MBC. The results would actually indicate that the MBC is higher for BA-NH-NH-BA (250mM) compared to butyrate (100mM).

Given the relatively high concentrations used, it would be important to test for cytotoxicity. This could be done with the HaCat cell line or by tissue analysis from mice treated with BA-NH-NH-BA. Would it be possible that the increase of IL-6 is due to tissue damage by BA-NH-NH-BA?

Fig. 6b-d: it would be interesting to see a comparison with equimolar amounts of butyrate. Where the wound sites infected with S. aureus first, before BA-NH-NH-BA was applied or where they pre-mixed?

Why was an AD isolate used? Would a S. aureus isolate from a healthy skin lead to the same results?

Line 226-229: this statement is highly speculative

Minor issues:

Lines 35/36: Some genera/species are shown in bold? Also, sentence in line 46. Any particular reason?

Line 78: 3 days seems a long time for growing S. aureus on MSA

Line 127-130: this should probably read “two bacterial species”

Fig. 6b: Should the p-value for 0.4mM be “*” (<0.05)?

Author Response

Summary

As reviewer requested, two new experiments in Figures S6 and   S7 have been conducted to demonstrate no cytotoxicity and high stability of   BA-NH-NH-BA. Furthermore, two new references [37, 38] were added.

Comment 12:

This manuscript describes the isolation of a S.   aureus strain from an atopic dermatitis lesion and the effect of   BA-NH-NH-BA on S. aureus growth, IL-6 production and histone   acetylation.

The effect of butyrate on S. aureus   growth, IL-6 production and histone acetylation has been described before in   the literature, as well as the association of S. aureus with AD. The   novelty of this manuscript lies in the generation of a water-soluble   derivative of butyrate, BA-NH-NH-BA. The authors suggest that this derivative   is more active than butyrate and might be useful as a therapy against AD.

The first three figures show data that   confirm results from previous studies (production of butyrate from glycerol   by S. epidermidis, effect on IL-6 levels and growth suppression of S.   aureus).

The section around figures 4 and 5 is   confusing. The authors describe “complete inhibition”, but what they actually   test is the minimal bactericidal concentration (MBC) which is lower than the   MBC. The MBC is defined as the concentration that kills at least 99.9% of a   106/ml CFU culture. In the method section and fig 4, it says 106/ml   CFU were used, fig 5 says 105/ml CFU and the Y-axes in both   figures indicate 108/ml CFU as starting point. Either way, a one   to two log reduction is not equivalent to MBC. The results would actually indicate   that the MBC is higher for BA-NH-NH-BA (250mM) compared to butyrate (100mM).

Response 12:

1.     The numbers (CFU) of AD S. aureus in Figures 4 and 5 used in this study have been   corrected.

2.     The MBC was defined as the lowest   concentration of substance, which produced ≥99.9% killing after 24 h of   incubation as compared to the colony count of the starting inoculum (Biomed   Res Int, 2015, 349534). We used the log10 to express the number of   AD. S. aureus. The 90%, 99% and   99.9% killing can be calculated as > 1 log10, 2 log 10,   and 3 log10, respectively. 

3.     To avoid the confusion, we have changed the term of “MBC” to “>1 log10 inhibition”.

“The >1 log10   inhibition of BA-NH-NH-BA for AD S. aureus was 0.02 mM, while   concentrations greater than 250 mM completely inhibited the growth (Figure 5b   and c).”

Comment 13: Given the relatively high   concentrations used, it would be important to test for cytotoxicity. This   could be done with the HaCat cell line or by tissue analysis from mice   treated with BA-NH-NH-BA. Would it be possible that the increase of IL-6 is   due to tissue damage by BA-NH-NH-BA?

Response 13:

A new experiment in Figure S6   has been conducted to demonstrate the low cytotoxicity of BA-NH-NH-BA to   mouse skin.

2.4. Terminal deoxynucleotidyl transferase dUTP nick end labeling (TUNEL)   Assay

To examine the cytotoxicity of   BA-NH-NH-BA, dorsal skin of ICR mice were topically applied with BA-NH-NH-BA   (4 mM) or PBS for 24 h. The skin was excised, immersed and fixed in 10%   formalin. The tissue sections of skin were cut with a thickness of 3 µm for TUNEL staining (R&D systems, Minneapolis, MN,   USA). To quantify the TUNEL-negative (nuclear, blue staining) and -positive   (nuclear, brown staining) cells, a total of at least 3 randomly selected   stained images with more than 50 cells were counted.”

“Figure S6. No significant   cytotoxic effect of BA-NH-NH-BA. (a) Histology (TUNEL staining) of mouse skin   (epidermis and dermis) 24 h after topical application of 4 mM BA-NH-NH-BA or   PBS. Scale bars: 30 µm. (b) Percentage of (TUNEL-negative) live cells in skin   applied with BA-NH-NH-BA or PBS was quantified. Data   shown are mean ± SE. ns, not significant.”

Comment 14:

Fig. 6b-d: it would be interesting to see a   comparison with equimolar amounts of butyrate. Where the wound sites infected   with S. aureus first, before BA-NH-NH-BA was applied or where they   pre-mixed?

Response 14:

The valuable comment has been   added into Discussion Section.

“To mimic the over-growth of S. aureus in AD lesions, future works   will include inoculation of AD S.   aureus onto skin for few days before topical application of BA-NH-NH-BA.”

Comment 15:

Why was an AD isolate used? Would a S.   aureus isolate from a healthy skin lead to the same results?

Response 15:

The valuable comment has been   added into Discussion Section. Two new references have been cited.

“It has been reported that AD S. aureus and other S.   aureus strains have different characteristics including distinct   activities of clumping factor B [37] and T cell responses [38]. Our future   works will determine if BA-NH-NH-BA can selectively suppress the growth of AD   S. aureus without affecting other   skin commensal bacteria.”

37.  Fleury, O. M.; McAleer, M.A;, Feuillie,C.;   Formosa-Dague, C.; Sansevere, E.; Bennett, D.E.; Towell, A.M.; McLean, W.I.;   Kezic, S.; Robinson D.A. Clumping factor B promotes adherence of   Staphylococcus aureus to corneocytes in atopic dermatitis. Infection and immunity 2017, 85, e00994-00916.

38.  Iwamoto, K.; Moriwaki, M.; Niitsu, Y.;   Saino, M.; Takahagi, S.; Hisatsune, J.; Sugai M.; Hide, M. Staphylococcus   aureus from atopic dermatitis skin alters cytokine production triggered by   monocyte-derived Langerhans cell. Journal   of dermatological science 2017,88,   271-279.

Comment 16:

Line 226-229: this statement is highly   speculative.

Response 16:

The statement has been   clarified. Reference 36 has been cited.

“Like the antibacterial activity of water soluble and   hydrophilic chitosan [36], the anti-S. aureus activity of BA-NH-NH-BA may   be the result of changes in the properties of plasma membrane permeability,   thus provoking internal osmotic imbalance and consequently inhibit the growth   of bacteria.”

36.  Raafat, D.; Sahl,   H.G. Chitosan and its antimicrobial potential–a critical literature survey. Microbial biotechnology 2009, 2, 186-201.

Comment 17:

Lines 35/36: Some genera/species are shown in   bold? Also, sentence in line 46. Any particular reason?

Response 17:

The errors have been corrected.  

Comment 18:

Line 78: 3 days seems a long time for growing   S. aureus on MSA.

Response 19:

Three days allow other non-S. aureus bacteria for growth.

Comment 20:

Line 127-130: this should probably read “two   bacterial species”.

Response 20:

We have changed “two bacteria”   to be “two bacterial species”.

Comment 21:

Fig. 6b: Should the p-value for 0.4 mM be “*”   (<0.05)?

Response 21:

The error has been corrected.

Round 2

Reviewer 2 Report

In general, this is a much improved version of the manuscript. There are still some minor issues.

There is still some confusion around bacterial inhibition and killing. As the authors plate the bacteria, there are indeed looking for killing, so it would be OK to write ">1log10 killing" (lines 141 and 152), just NOT use the term "MBC" (line 145) which would require >3log10 killing.

The Y-axes in figures 4B and 5B are still wrong. The CFU/ml are higher than the starting culture.

Author Response

Comment 1:

In general, this is a much improved version of the manuscript. There are still some minor issues.

There is still some confusion around bacterial inhibition and killing. As the authors plate the bacteria, there are indeed looking for killing, so it would be OK to write ">1log10   killing" (lines 141 and 152), just NOT use the term "MBC"   (line 145) which would require >3log10 killing.

The Y-axes in figures 4B and 5B are still wrong. The CFU/ml are higher than the starting culture.

Response 1:

We have changed the words of “MBC value” (line 145)   to “>1 log10 inhibition”.

Y-axes in both figures indicated   108 CFU/ml are correct.

106 CFU /ml bacteria were used for experiment start times; bacterial numbers higher than 106 CFU /ml were detected after 24 h incubation.
